



# High-Resolution Urban Observation Network for a User-Specific Meteorological Information Service in the Seoul Metropolitan Area, Korea

Moon-Soo Park[1], Sung-Hwa Park[1], Jung-Hoon Chae[1], Min-Hyeok Choi[1], Yunyoung Song[1], Minsoo Kang[1]

[1]Weather Information Service Engine, Hankuk University of Foreign Studies, 17035

*Correspondence to*: Moon-Soo Park (ngeograph2@gmail.com)

**Abstract.** To improve our knowledge of urban meteorology, including those processes applicable to high-resolution meteorological models in the Seoul Metropolitan Area, a Weather Information Service Engine urban meteorological observation system network (UMS-Seoul) has been designed and installed. The network incorporates 14 surface energy balance (EB) systems, 7 surface-based 3-dimensional meteorological observation (3D) systems, and applied meteorological observation (AP) systems, as well as the existing surface-based meteorological observation network. The EB system consists of a radiation balance system, sonic anemometers, infrared $CO_2/H_2O$ gas analyzers, and many sensors to measure wind speed and direction, temperature and humidity, precipitation, and air pressure, etc. The EB-produced radiation, meteorological, and turbulence data will be used to quantify the surface energy balance according to land use, and improve the boundary layer and surface processes in meteorological models. The 3D system, composed of wind lidar, a microwave radiometer, an aerosol lidar, or a ceilometer, produces vertical profiles of backscatter by aerosols or water vapor, cloud height, wind speed and direction, temperature, humidity, and liquid water content. It will be used for high-resolution reanalysis data based on observations as well as for improvement of the boundary layer, radiation, and microphysics processes in meteorological models. The AP system includes road weather information, mosquito activity, and water quality observation instruments. The standardized metadata for networks and stations are documented and renewed periodically to provide a detailed observation environment. The UMS-Seoul data are designed to support real-time acquisition, as well as display and automatically quality check the data within 10 minutes of observation. After the quality check, data can be distributed to relevant potential users such as researchers and policy makers.

## 1 Introduction

The world population exceeded 7.3 billion in 2015 and is projected to increase steadily and reach 9.7 billion by 2050 and 11.2 billion by 2100 (United Nations, 2015). Urban population has also increased and is expected to increase at an even greater rate. The ratio of the global urban population was over 53 % in 2014 and is projected to grow to approximately 66 % by 2050 (United Nations, 2014). High population density in urban areas are inevitably vulnerable not only to disastrous



meteorological and environmental phenomena such as heavy rain/snow falls, heat and cold waves, air pollution, or strong wind, but also to man-made disasters such as the explosion or release of toxic gases (OFCM, 2004; Razafindrabe et al., 2009). Impervious surfaces in urban areas tend to amplify urban flash flooding under heavy rainfall conditions; freezing rain or snowfall disrupts transportation systems; and severe storms with lightning and high winds may result in power failures. A

5 high population density in urban areas therefore results in greater property damage and loss of life as a result of disastrous events.

It is well known that urban structure and morphology affect meteorology in various ways, including: an increase in temperature, leading to the urban heat island effect (Bornstein, 1968; Oke, 1973; Landsberg, 1981; Arnfield, 2003; Kalnay and Cai, 2003; Kim and Baik, 2005; Grimmond, 2006); a decrease or increase in the temporal variation of absolute humidity

due to impervious surfaces and anthropogenic water use (Unger, 1999; Kuttler et al., 2007); increased haze, cloud, and precipitation (Bornstein and Lin, 2000; Dixon and Mote, 2003; Shepherd, 2005; Carrio et al., 2010); decreased visibility due to anthropogenic aerosols (Cheng and Tsai, 2000; Singh et al., 2008; Nichol et al., 2010); increased turbulent intensity and changed wind speed due to high-rise buildings (Roth, 2000; Arnfield, 2003; Grimmond et al., 2004; Barlow et al., 2011; Song et al., 2013); and a decrease in solar radiation, resulting in increased sensible heat flux and heat storage (Nunez and

Oke, 1977; Christen and Vogt, 2004; Harman and Belcher, 2006; Grimmond et al., 2009; Nordbo et al., 2012; Park et al., 2014). The stronger the synoptic wind, the more the maximum upward motion and precipitation area moves downwind (Bornstein and Lin, 2000; Lin et al., 2011; Han et al., 2014).

Many countries and cities in Europe, North America, and Asia have conducted urban meteorological experiments and/or intensive observation campaigns for various purposes such as understanding urban meteorological processes, and improving

the predictability of urban high-resolution meteorological phenomena and user-specific meteorological information (Allwine et al., 2002; Cros et al., 2004; Rotach et al., 2005; Schroeder et al, 2010; Basara et al., 2011; Koskinen et al., 2011; Hicks et al., 2012; Wood et al., 2013; Nakatani et al., 2015; Tan et al., 2015). For instance, the BUBBLE (Basel Urban Boundary Layer Experiment) project conducted in the Basel area in Switzerland is famous for below and above street canyon observation using meteorological towers, which contributes to the development and verification of the street canyon type

computational fluid dynamics (CFD) model and urban canopy schemes in meteorological models. This project also includes flux measurements according to land cover (urban, suburban, and rural), wind lidar and sodar, and RASS observations (Rotach et al., 2005). Secondly, TOMACS (Tokyo Metropolitan Area Convection Study for Extreme Weather Resilience Cities) focuses on severe weather processes such as severe storms and typhoons. Many kinds of radars, including Ku-band, X-band and C-band polarimetric radar, as well as disdrometer, wind lidar, microwave radiometer, and scintillometers, have

been installed in the Tokyo metropolitan area as part of the TOMACS project (Nakatani et al., 2013; 2015). Thirdly, SUIMON (Shanghai Urban Integrated Meteorological Observation Network) in Shanghai in China aims to forecast and service meteorological information on macro-, meso-, urban-, neighborhood-, and street- scales using the platform of radar, wind profilers, ground-based and satellite-based remote sensing, and in-situ observations (Tan et al., 2015).



For the purpose of a high-resolution meteorological information service, improvement of the supporting meteorological model is essential. Physics schemes, including microphysics, cumulus, radiation, surface, and atmospheric boundary layer models, all interact with each other (Dudhia, 1989). Irregular surface morphology and surface materials in urban areas affect the surface optical, physical, and thermal properties such as thermal conductivity, heat capacity, roughness, displacement

length, albedo, and emissivity (Masson, 2006; Lee and Park, 2008; Grimmond et al., 2009). These properties change the energy partition dramatically over urban surfaces compared to rural surfaces. The resulting sensible and latent heat fluxes alter the boundary-layer structure through interactions between surface, radiation, and boundary layer processes (Pielke, 2002).

In Korea, the urban population ratio was 82.4 % in 2014 and has increased steadily from 21.4 % in 1950, to 79.4 % in 2000,

and is expected to reach 87.6 % in 2050 (United Nations, 2014; 2015). The Seoul Metropolitan Area (SMA) was ranked as the fifth largest urban area population in 2015 (Demographia, 2015). Meteorological data analyses for the period from 1960 to 2009 in SMA show that, in this area, air temperature and precipitation increase, relative humidity decreases, and heavy rainfall events with more than 20 mm h$^{-1}$ also increase (Kim et al., 2011). Recently, SMA has experienced: blackouts of more than 1.6 million houses due to failure in electric power demand prediction after unreasonably hot weather in autumn;

massive damage from shallow landslides due to heavy rainfall in 2011 (Park et al., 2013a); several inundations of urban centers by flash floods (Kim et al., 2014); building damage from strong winds such as typhoons; traffic accidents as a result of road ice; and deaths from heat/cold waves every year (Son et al., 2012).

The Weather Information Service Engine (WISE) project was launched in 2012 to meet the needs of user-specific high-resolution meteorological information in order to reduce damage to the SMA caused by extreme weather phenomena (Choi

et al., 2013). In order to achieve these goals, scientific advances in urban meteorology and development/improvement of high-resolution meteorological models and service specific application models are needed (Baklanov, 2006; Baklanov et al., 2008). To service the observation-based meteorological information and support the development or improvement of related models, a high-resolution urban meteorological observation system network has been proposed and established in SMA (UMS-Seoul).

This study includes the background of UMS-Seoul through a description of the geography, topography, and land cover of the SMA, and a review of the existing available meteorological observation networks. Then, we present the objectives, details, and applications of each meteorological observation system network including the surface energy balance observation system, the 3-dimensional meteorological observation system, and the applied meteorological observation system network in UMS-Seoul.

**2 Seoul Metropolitan Area**

The Seoul Metropolitan Area (SMA) on the Korean peninsula consists of three administrative provinces: Seoul Special City, Incheon Metropolitan City, and Gyeonggi Province (Fig. 1a). Seoul Special City, the capital city of Korea, is surrounded by



Gyeonggi Province and Incheon Metropolitan City, with the highest population density of 16,188 km$^{-2}$ (Table 1). Incheon Metropolitan City is located between Seoul Special City and the Yellow Sea. Gyeonggi Province has the highest population of 11.4 million and the largest area of 10,184 km$^2$, but the lowest population density of 1,119 km$^{-2}$ (Table 1).

SMA has very complex geography, topography, and land cover. To the west of SMA is the Gyeonggi Bai in the Yellow Sea, whose coastline is very irregular. The Yellow Sea is often an important moisture source in the case of heavy rainfall, snowfall, or heterogeneous reactions among long-range transporting air pollutants (Chung and Kim, 2008; Cayetano et al., 2011; Cha et al., 2011; Park et al., 2011; 2013b; Jeong and Park, 2013). The western part of SMA is relatively low-lying farmland or urban areas, while the eastern part is high-altitude mountain ranges, some of which are higher than 1,000 m in Domain 1 (Fig. 1a). Most mountains in Korea are covered by forest. High-populated areas range from Incheon Metropolitan City to Seoul Special City, indicated in Domain 2 (Fig. 1b). The Han River flows from east to west, and divides SMA as well as Seoul Capital City. Seoul Capital City is surrounded by several high mountains above 500 m in altitude: Bukhan, Dobong, Surakn, and Gwanak mountains in the northern part; and Cheonggye and Bulam mountains in the southern part. There is a small mountain (262 m high) in the center of Seoul Special city (Fig. 1b).

Figure 2 shows the simplified land use in SMA (Domain 1) and high-populated areas (Domain 2) with 90 m horizontal resolution. Forest covers 41.9 % of the area, croplands including pasture and grassland cover 21.5 %, water bodies including seawater as well as inland water cover 20.9 %, urban areas including residential, industrial, and commercial areas cover 8.6 %, and wetlands cover 5.0 % in Domain 1 (Table 2). In Domain 2, forest covers 36.0 %, urban covers 28.3 %, croplands cover 20.6 %, wetlands cover 8.6 %, and water bodies cover 6.4 % of the area (Table 2). Most wetlands are tidelands on the border between the continent and the Yellow Sea. More than 40 % of Seoul Capital City is covered by residential or commercial areas, approximately 30 % is covered by forest, and around 10 % is covered by roads and rivers (Fig. 2b).

## 3 Background Meteorological Observation Network

Many background surface meteorological observation systems have already been installed in SMA (Fig. 3). There are over 110 Korea Meteorological Administration (KMA)-operated meteorological observation stations, whose locations are selected based on administrative district, and more than 1,000 SKP (SK Planet) private telecommunication company-operated meteorological observation stations, whose locations are selected based on floating and resident populations in Domain 1. The KMA operated automatic synoptic observation system (ASOS) observes the air pressure, evaporation, cloud amount, sunshine, snow depth, surface and ground temperatures, and weather phenomena, as well as the basic meteorological variables deployed on the grass surface with minimized obstacles. The station location of ASOS follows the guidelines of the World Meteorological Organization (WMO, 2008). The KMA operated automatic weather system (AWS) has a wind speed/direction sensor at 7 or 10 m, a temperature sensor at 1.5~2 m, precipitation detection, and a tipping-bucket type rain gauge with heater, while each SKP operated AWS has an integrated meteorological sensor and tipping-bucket type rain gauge without heater, which is set to do not measure precipitation in winter. Some KMA operated AWS and most SKP



operated meteorological stations are installed on the rooftops of buildings or in street canyons between buildings. Regardless of installation environment, if we choose any point with urban land cover in Seoul Special City, the distance from that point to the nearest AWS will be less than 1 km.

There are 2 rawinsonde stations, 2 wind profiler and microwave radiometer stations, and 6 radar stations in Domain 1 (Fig. 3). Osan (WMO Station Number 47122) and Baengnyeongdo (WMO station number 47102) stations observe upper air meteorological variables 4 and 2 times a day, respectively. And Paju and Cheorwon stations installed a wind profiler and microwave radiometer to observe the vertical profile of wind, temperature, and humidity. Regarding radar stations, KMA operates 3 S-band and 1 C-band radar, the Korea Institute of Civil Engineering operates 1 X-band radar, the Korea Air Force operates 1 C-band radar, and the Ministry of Land, Infrastructure, and Transport operates 1 C-band radar in Domain 1 (Fig. 3).

Even though high-resolution and various meteorological observation systems are located in SMA, there are still many unknowns regarding urban surface forcing and vertical profiles of temperature, humidity, and wind that impede our understanding of the fundamentals of urban meteorological phenomena in highly populated areas of SMA. To counter this, the WISE project has designed and installed the UMS-Seoul in this area.

## 4 Urban meteorological observation network and its applications (UMS-Seoul)

UMS-Seoul is composed of a surface energy balance observation network (EB), a 3D meteorological observation network (3D), and an applied meteorological observation network (AP), as well as the existing surface-based meteorological observation network in SMA (Table 3).

Fig. 4 shows the location of each meteorological observation network station in UMS-Seoul. Station locations are selected considering the surface land cover and the horizontal distribution of geography and topography. Table 4 shows the land cover of major observation stations classified by Auer (1978), Davenport et al. (2000), Oke (2004), and Stewart and Oke (2009; 2012). Each station represents a different type of land cover. Yeouido and Gwangjin stations, located on the border of the Han River, are representative river sites. Typifying intensely developed and compact high-rise sites, Gwanghwamun, Guro, and Songdo are located at the center of Seoul Special City, the southwest of Seoul Special City, and the south of Incheon Metropolitan City, respectively. Songdo station, in particular, is a newly developed high-rise building complex. Anyang and Nowon stations are surrounded by apartment building complexes, and are therefore representative residential housing areas. Jungnang, Gajwa, Gangnam, and Seongnam stations are located in representative residential areas in Seoul Special City and Gyeonggi Province. As rural stations, Youngin and Bucheon stations represent urban forest and crop field sites.

All 14 surface energy balance observation (EB) systems are installed on the ground, or on the rooftops of buildings over the representative surface land covers (residential, commercial, industrial, mixed, and rural). Measurement tower height ranges from 1.5 m (river side) to 18.5 m (Jungnang). Each system includes 2 or 3 temperature, relative humidity, wind speed, and



wind direction sensors, 1 or 2 $CO_2/H_2O$ infrared gas analyzers, 2 or 3 sonic anemometers, an air pressure sensor, a rain gauge with heater, a surface temperature sensor, and 4 components of net radiometers, i.e., downward/upward and shortwave/longwave radiometers (Fig. 6). Additionally, a large aperture scintillometer and thermal infrared imagery systems are installed to obtain the line-averaged sensible heat flux and sub-building scale spatial distributions of surface temperature.

EB systems installed on different land cover in urban areas determine not only the surface thermal, optical, and physical properties such as thermal conductivity, heat capacity, albedo, emissivity, roughness length, and displacement length, but also the surface energy balance among net radiation, sensible heat flux, latent heat flux, heat storage or ground heat flux, and anthropogenic heat flux. They are expected to produce high-resolution surface property maps, such as albedo, emissivity, and thermal conductivity, as well as surface roughness length and displacement. Furthermore, they determine the 30-minute

averaged carbon dioxide concentration and flux as well as the sensible heat flux, the latent heat flux, the radiative flux, and heat storage. EB data are applied to verify the urban surface processes according to land use, and improve the urban surface processes in the model. Fig. 5 shows a typical EB measurement tower including sensors, representative surface land cover, and an example of a time series of wind speed, temperature, friction velocity, and sensible heat flux at an urban residential area in Seoul Special City.

The 3D meteorological observation (3D) network gives the real-time vertical profile of backscatter, wind speed and direction, temperature, and humidity using the 2 aerosol lidars, 2 ceilometers, 7 wind lidars, and 7 microwave radiometers. Aerosol lidar gives the vertical distribution of aerosols and aerosol optical depth using the vertical profile of range-corrected backscatter signal and depolarization ratio by 532 nm and 1064 nm wavelength lasers. The ceilometer gives the vertical distribution of water vapor and cloud base height using the vertical profile of backscatter by a 910 nm wavelength laser.

Wind lidar gives the vertical profile of wind speed and direction using the Doppler shift by a 1532 nm wavelength laser. Microwave radiometer gives the vertical profile of temperature and humidity using the observed brightness temperature of 14 wavelengths. Figure 6 shows images of installed instruments and examples of vertical backscatter profiles of aerosols and water vapor (ceilometer, aerosol lidar), temperature and relative humidity (microwave radiometer), and wind speed and direction (wind lidar).

The Applied Meteorological Observation (AP) system includes road weather, water quality, mosquito, and agro-meteorology information systems. Each AP system is designed to support each applied information service through the verification and improvement of each information service-capable model. For example, the Road Weather Information System (RWIS) observed the road surface temperature and status, water depth, salinity, and conductivity, as well as net radiation, temperature, humidity, wind speed, and wind direction on an open road in 2013, at the entrance and exit of a tunnel in 2014,

and on a complex road structure with bridge and joint cross section in 2015. These data will be used to verify and improve the road surface temperature, status, and braking distance prediction model (Park et al., 2014). The water quality information system observes the turbidity, salinity, conductivity of stream water, dissolved oxygen, and biological oxygen demand in order to support water quality information. The mosquito activity information system observes the number of mosquitoes and meteorological variables for the purpose of a mosquito advisory and warning service. The agro-meteorological



information system observes the soil moisture and temperature, ground heat flux, evaporation, leaf wetness, and leaf temperature, as well as basic meteorological variables including sensible and latent heat fluxes for predicting the productivity of crops. Figure 7 shows the installed AP systems and examples of observed data.

## 5 Data Processes

### 5.1 Metadata

In order to understand the environment surrounding the sites in more detail, and maintain the networks and sites more efficiently, metadata for UMS-Seoul is standardized by a comparison with data established by the World Meteorological Organization, the Korea Meteorological Administration, and previous studies (WMO, 2008; KMA, 2013; Muller et al., 2013; Song et al., 2014). The UMS-Seoul metadata includes network and station information composed of general, local-scale, micro-scale, and visual information (Muller et al., 2013). Figure 8 shows the structure of WISE metadata and an example of station general information metadata (Song et al., 2014). The metadata contains static information such as urban structure, surface cover, metabolism, communication, building density, roof type, moisture/heat sources, and traffic, as well as updated information on environmental changes, maintenance, replacement, and/or calibration of sensors. Network and station metadata are required to be documented and updated every year.

### 5.2 Data acquisition and display system

All data are collected, displayed, and quality checked in real-time (Fig. 9). When data are sampled on the 00 minute of every hour, they are transmitted into a server at 05 minutes using machine to machine (M2M) technology by code division multiple access (CDMA) or a long term evolution (LTE) communication network, and the subsequent automated quality checks are conducted in 10 minutes. Then, the data are ready to distribute to the relevant users. WISE-Mnet quality checks are divided into automated steps and manual steps. Basic quality checks include missing checks, physical limit checks, climate range checks, and spike removal (Chae et al., 2014; Kwon et al., 2014; Park and Choi, 2016).

Not only are the current values and time series for automated quality-checked data displayed in real time, but also the derived variables such as albedo, atmospheric boundary layer height, net radiative flux, etc. in order to monitor the past and current state of atmospheric variables (Fig. 9).

## 6 Summary and Discussion

The UMS-Seoul is one of the most intensively integrated urban observation networks in the world for user-specific meteorological information such as flash floods, road status and surface temperature, urban heat islands, air quality, etc. due to retrieval of high-resolution and high-quality meteorological data from the Seoul Metropolitan Area in Korea.



Although the existing surface meteorological observation network provides very high-resolution information, it focuses primarily on surface meteorology and radar echoes. The UMS-Seoul includes an additional 14 surface energy balance systems, 2 aerosol lidar systems and a ceilometer, 7 wind lidar systems and microwave radiometers, and applied meteorological observation systems. The surface energy balance system determines surface energy forcing as well as

physical, optical, and thermal properties according to surface land use. Wind lidars and microwave radiometers provide vertical profiles of wind, temperature, humidity, and liquid water in real-time. The applied meteorological observation system includes road, water quality, greenhouse gas, and mosquito data in order to improve the user-specific weather information service.

Metadata are standardized to give detailed network and station information. Also, real-time data acquisition, quality checks,

and data display are constructed to monitor the horizontal and vertical distribution of meteorological variables. Qualified data are then ready to distribute to researchers or policy makers.

In addition to the fixed meteorological observation systems, a mobile road weather and meteorological observation system can be installed on a car. It ascertains the road slope angle and surface material properties. Through several intensive upper-air observation experimental campaigns at SMA, we aim to have a deeper comprehension of urban meteorology and produce

the data necessary for a high-resolution meteorological information service. After comprehensive analysis of the sampled data and the observing system simulation experiment (OSSE) is conducted, the optimal locations of instruments are re-selected. Following the analysis results, instruments can be inserted, removed, or moved to another stations.

As a leading urban experimental complex, UMS-Seoul is expected to contribute to the improvement of high-resolution meteorological information technology and the alleviation of damage from disastrous weather phenomena in high-population

density urban areas around the world.

**Acknowledgements**

This work was funded by the Weather Information Service Engine (WISE) Program of the Korea Meteorological Administration under Grant KMIPA-2012-0001-01. All data used in this study were also produced under the WISE Program.

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

**Table 1 Area, population, and population density statistics in SMA (http://kosis.kr).**

|  | Seoul | Incheon | Gyeonggi |
|---|---|---|---|
| Area (km$^2$) | 605 | 1,010 | 10,184 |
| Population (thousands) | 9,794,304 | 2,662,509 | 11,379,459 |
| Population density (km$^{-2}$) | 16,188 | 2,588 | 1,119 |

**Table 2 Percentage of land use in the SMA domain (Domain 1) and the high-populated domain (Domain 2).**

| Land use | Domain 1 (%) | Domain 2 (%) |
|---|---|---|
| Cropland, pasture, grassland | 21.5 | 20.6 |
| Forest | 41.9 | 36.0 |
| Water bodies | 20.9 | 6.4 |
| Wetlands | 5.0 | 8.6 |
| Barren | 2.1 | 0.1 |
| Urban (low intensity residential) | 4.3 | 12.8 |
| Urban (high intensity residential) | 2.2 | 9.0 |
| Urban (industrial & commercial) | 2.0 | 6.5 |



**Table 3 Specification and installed sensors of the existing meteorological observation network in the Seoul Metropolitan Area**

| Systems | Descriptions | | | |
|---|---|---|---|---|
| Automatic Synoptic Observation System | Operator | Korea Meteorological Administration | | |
| | Number | 7 | | |
| | Variable | Temperature, relative humidity, wind speed, wind direction, solar radiation, grass temperature, underground temperature, air pressure, evaporation, sunshine, snow depth, precipitation, precipitation detection, (manual) weather phenomena, cloud amount | | |
| Automatic Weather System | Operator | Korea Meteorological Administration | | |
| | Number | 108 | | |
| | Sensors | Temperature, wind speed, wind direction, air pressure, precipitation, precipitation detection | | |
| Integrated Meteorological Observation Station | Operator | SK Planet | | |
| | Number | 1078 | | |
| | Sensors | Temperature, relative humidity, wind speed, wind direction, air pressure, precipitation | | |
| Radar | Operators | Korea Meteorological Administration | Ministry of Land, Infrastructure and Transport | Korea Air Force | Korea Institute of Civil Engineering |
| | Number | 4 | 1 | 1 | 1 |
| | Specifications | S-band 3 C-band 1 | C-band | C-band | X-band |
| Wind profiler | Operator | Korea Meteorological Administration | | |
| | Number | 2 | | |
| | Specifications | UHF (Ultra High Frequency) 1.29 GHz | | |
| Radiometer | Operator | Korea Meteorological Administration | | |
| | Number | 2 | | |
| | Specification | Humidity 7 channel (22~30 GHz), Temperature 7 channel (51~59 GHz) | | |
| Rawinsonde | Operator | Korea Meteorological Administration | | |
| | Number | 2 | | |
| | Specification | 00, 12 UTC for regular (06, 18 UTC option) | | |



**Table 4 Land cover classification of major observation stations.**

| Station | | Classification | | | | |
|---|---|---|---|---|---|---|
| Name | ID | Auer (1978)[1] | Davenport et al. (2000)[2] | Oke (2004)[3] | Stewart and Oke (2009) | Stewart and Oke (2012)[4] |
| Jungnang | 02201 | R2 | N7 E7 S7 W7 | UCZ-2 | Compact housing | LCZ-2E |
| Gwanghwamun | 02202 | C1 | N8 E8 S8 W8 | UCZ-1 | Modern core | LCZ-1E |
| Myeonmok | 02203 | R2 | N7 E7 S7 W7 | UCZ-2 | Compact housing | LCZ-2E |
| Gajwa | 02205 | R2 | N7 E7 S7 W7 | UCZ-3 | Compact housing | LCZ-3E |
| Guro | 02206 | C1 | N8 E8 S7 W6 | UCZ-1 | Modern core | LCZ-1E |
| Anyang | 02207 | R1 | N7 E7 S7 W7 | UCZ-3 | Blocks | LCZ-4B |
| Yeouido | 02208 | A5 | N1 E1 S7 W1 | UCZ-6 | Open ground | LCZ-4G |
| Bucheon | 02209 | A2 | N3 E4 S3 W3 | UCZ-7 | Cropped fields | LCZ-9D |
| Ilsan | 02210 | A2 | N5 E4 S4 W4 | UCZ-7 | Cropped fields | LCZ-9D |
| Songdo | 02211 | A1 | N6 E6 S6 W6 | UCZ-7 | Modern core | LCZ-4E |
| Youngin | 02212 | A1 | N7 E7 S7 W7 | UCZ-6 | Forest | LCZ-9A |
| Nowon | 02213 | R1 | N5 E5 S5 W5 | UCZ-3 | Blocks | LCZ-4B |
| Gangnam | 02214 | R2 | N7 E7 S8 W7 | UCZ-3 | Compact housing | LCZ-3E |
| Seongnam | 02215 | R2 | N7 E7 S7 W7 | UCZ-3 | Compact housing | LCZ-3E |
| Gwangjin | 02216 | A5 | N7 E1 S1 W1 | UCZ-6 | Open ground | LCZ-4G |

[1]Auer (1978): A1 metropolitan natural, A5 water surfaces, C1 commercial, R1 common residential, R2 compact residential.

[2]Davenport et al. (2000): N north, E east, S south, W west; 1 sea, 3 open, 4 roughly open, 5 rough, 6 very rough, 7 skimming, 8 chaotic.

[3]Oke (2004): UCZ (urban climate zone)-1 intensely developed, UCZ-2 intensely developed high-density, UCZ-3 highly developed medium density, UCZ-6 mixed use with large buildings in open landscape, UCZ-7 semi-rural development with scattered houses.

[4]Stewart and Oke (2012) LCZ (local climate zone) 1 compact high-rise, 2 compact midrise, 3 compact low-rise, 4 open high-rise 9 sparsely built; A dense trees, B scattered trees, C bush scrub, D low plant, E bare rock or paved, G water.



**Table 5 Specification and installed sensors of the surface energy balance system, 3D meteorological observation system, and applied meteorological observation system.**

| Systems | | Sensor or specification |
|---|---|---|
| Surface energy balance system | Sites | 14 (rural, residential, commercial, industrial, apartment, river) |
| | Tower | 1.5 m ~ 18.5 m |
| | Sensor | Temperature, relative humidity, wind speed, wind direction, downward/upward or shortwave/longwave radiation, $CO_2/H_2O$ infrared gas analyzer, sonic anemometer, surface temperature, rain gauge, water temperature (2 stations only), infrared thermometry (6 stations only) |
| | Option | Large Aperture Scintillometer 1 set<br>Surface temperature monitoring system 2 sets |
| 3D meteorological observation system | Ceilometer | - 2 stations<br>- Wavelength: 910 nm<br>- Backscatter by aerosol (up to 15 km, 10 m vertical resolution, 1 minute temporal resolution), cloud bottom heights (3 level) |
| | Aerosol lidar | - 2 stations<br>- Wavelength: 532 nm (parallel, cross-polarized), 1064 nm<br>- Backscatter by aerosol (up to 16 km, 3.75 m vertical resolution, 1 hour resolution), depolarization ratio, backscatter |
| | Radiometer | - 7 stations<br>- Water vapor (22~31 GHz, 7 channels), temperature (51~58 GHz, 7 channels)<br>- Brightness temperature for each channel, vertical profile of temperature, humidity, liquid water content |
| | Wind lidar | - 7 stations<br>- Wavelength: 1532 nm<br>- Wind speed and direction (up to 6,000 m, 100 m vertical resolution, 10 minute interval) |
| Applied meteorological observation system | Road | - 6 stations<br>- Wind speed and direction, temperature, humidity, precipitation, precipitation detection, insolation, net radiometer, road temperature and status, salinity, water depth |
| | Water quality | - 2 stations<br>- Water temperature, pH, conductivity, dissolved oxygen, salinity, turbidity, chlorophyll-a, water depth |
| | Mosquito | - 3 stations<br>- Mosquito activity |
| | Greenhouse gas | - 1 station<br>- $CH_4$ concentration, total radiation, diffuse radiation |
| | Agro-meteorology | - 4 stations<br>- Shortwave/longwave radiation, temperature and humidity, albedo, leaf wetness, soil moisture content, wind speed and direction, precipitation, soil temperature |





**Figure 1: (a) Geography and topography of the Seoul Metropolitan Area with administrative boundaries. (b) Enhanced geography and topography with major mountains in a highly populated region shown by the rectangle in (a).**







**Figure 2: (a) Land use in the Seoul Metropolitan Area, and (b) zoomed image of the highly populated region shown by the rectangle in (a).**



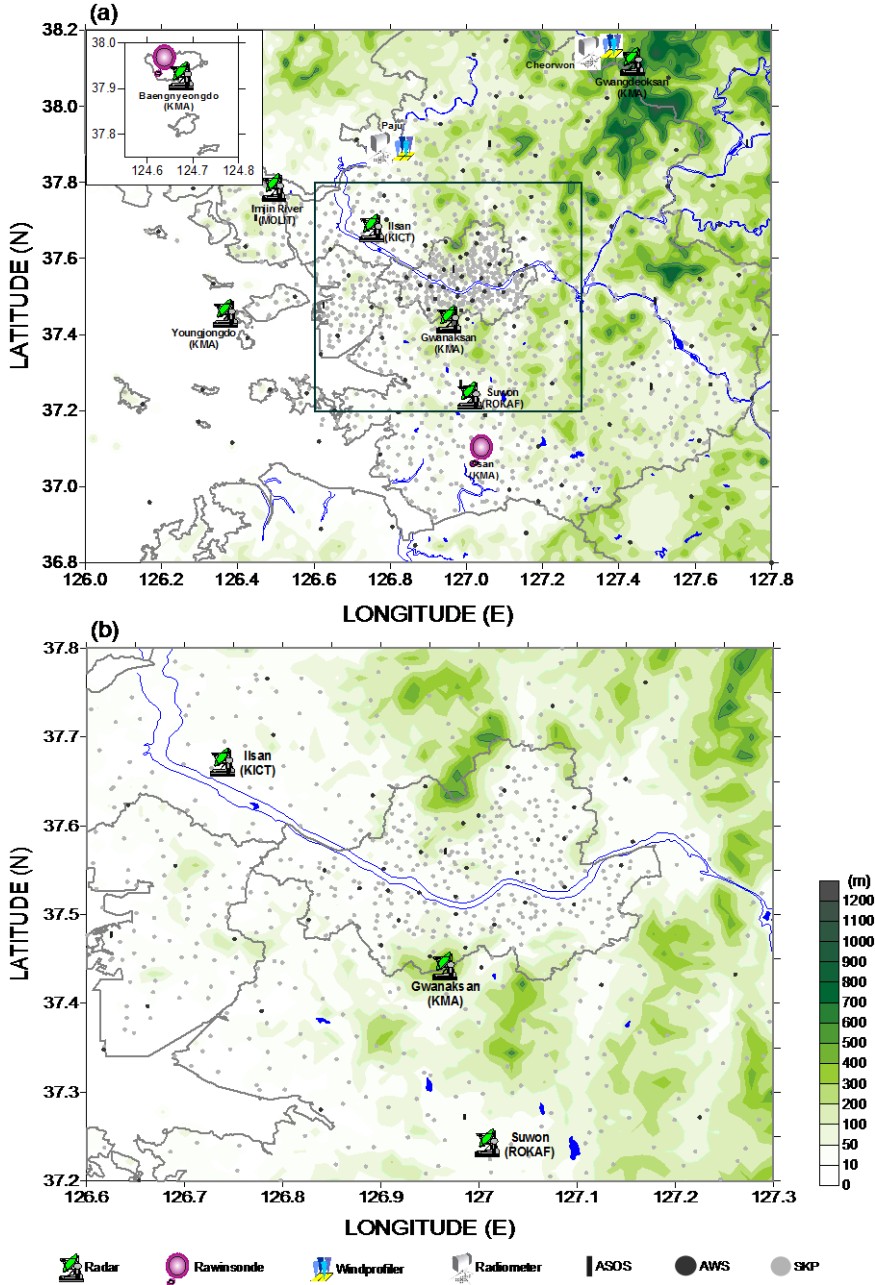

**Figure 3: Location of background radars, rawinsondes, wind profilers, radiometers, automatic synoptic observation systems, automatic weather systems, and SK-planet automatic weather systems in (a) the Seoul Metropolitan Area and (b) the highly populated region.**




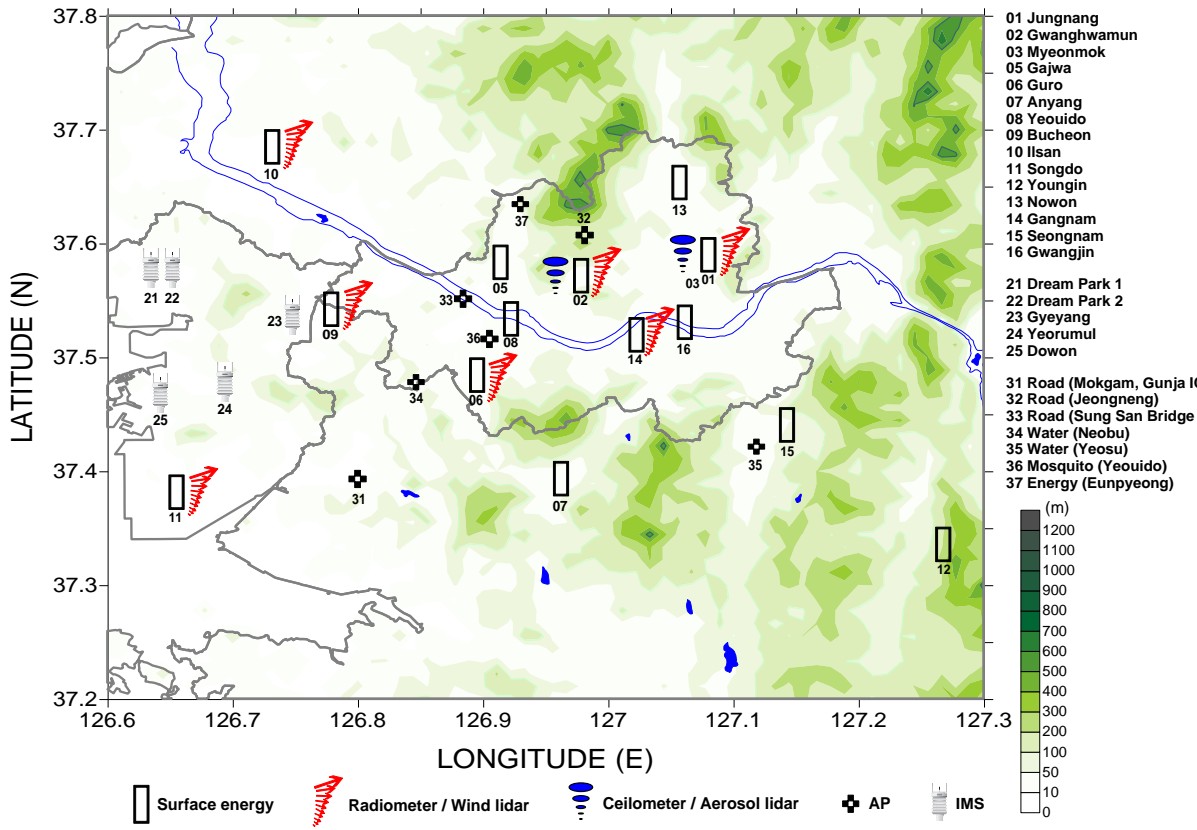

**Figure 4: Location of the UMS-Seoul urban meteorological observation system networks in the Seoul Metropolitan Area.**





Figure 5: (a) Sensor deployment of a typical surface energy balance system, and representative surface cover of (b-1) residential, (b-2) apartment, (b-3) industrial area, (b-4) urban rice paddy, (b-5) forest, and (b-6) river areas.



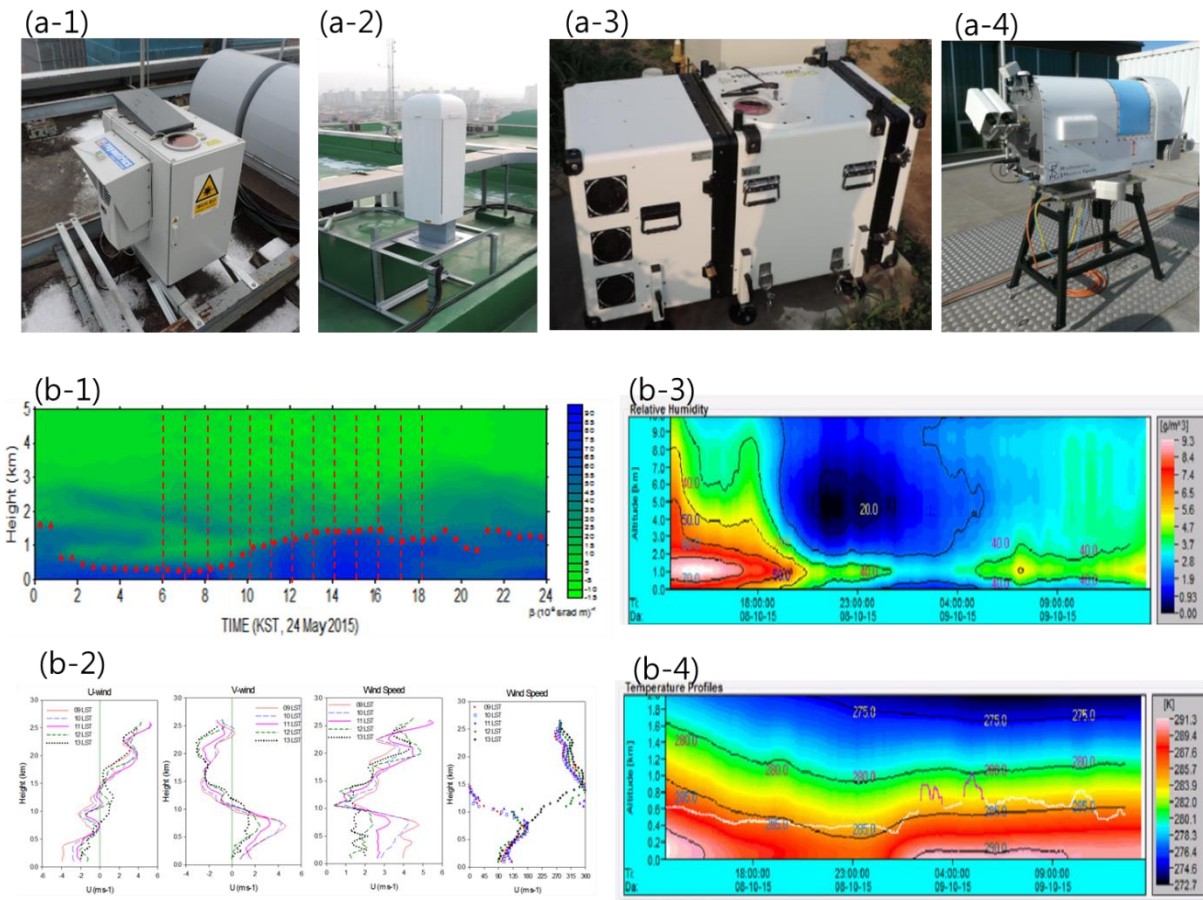

**Figure 6: Images of installed (a-1) aerosol lidar, (a-2) ceilometer, (a-3) wind lidar, and (a-4) microwave radiometer, and examples of vertical profiles of (b-1) backscatter by aerosols with an estimated atmospheric boundary layer height observed by the ceilometer, (b-2) u-wind, v-wind, wind speed, and wind direction observed by wind lidar, (b-3) relative humidity, and (b-4) temperature observed by microwave radiometer.**



**Figure 7: Images of (a-1) road weather information observation system, (a-2) agro-meteorological observation system, (a-3) mosquito activity observation system, (a-4) water quality monitoring system, and examples of times series of (b-1) road surface temperature at the entrance (red) and exit (blue) of a tunnel, (b-2) monthly mean number of mosquitoes, and (b-3) water quality including precipitation, water level, water temperature, dissolved oxygen, and dissolved oxygen saturation.**







**Figure 8: (a) The structure of UMS-Seoul metadata and station metadata, and (b) an example of station general information metadata for the surface energy balance system at Jungnang station.**





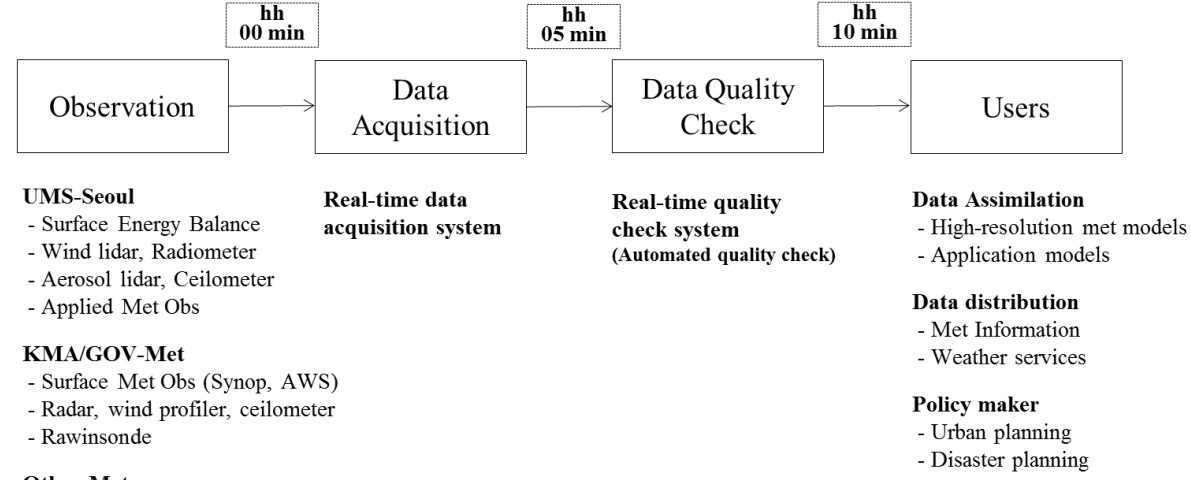

**Figure 9: UMS-Seoul data flow from observation to users via data acquisition and quality checks.**