# Peer review of "High-Resolution Urban Observation Network for a User-Specific Meteorological Information Service in the Seoul Metropolitan Area, Korea"

_Atmospheric Measurement Techniques, 2016_

## Referee Comment (RC1) · Anonymous Referee #1 · 31 Oct 2016

This paper introduces an extensive meteorological measurement network to support urban climate studies in the Metropolitan area of Seoul, Korea. While observation networks that offer high spatial coverage of a variety of atmospheric variables are important for an improved understanding of urban climate conditions, the presented work is not suitable for publication in AMT. It does not present any advancement in atmospheric measurement techniques or methodology but rather lists the different types of sensors being deployed. Of course, monitoring a variety of atmospheric and surface conditions at high spatial and temporal resolution offers great potential for integrated analysis and model comparison. However, no analysis is presented that demonstrates a new approach. The paper does not fit the scope of the AMT journal, however, given
the measurement network is very impressive and addresses the great need for detailed observations in urban areas, I would encourage the authors to submit the paper elsewhere.

Further, they should take into account the following aspects:

Text, maps, pictures should include much more details on the observing systems. For example, for all sensors, manufacturer and model should be listed. How do sensors inter-compare? How do you ensure absolute calibration? What is the accuracy of different measurements. What is the temporal resolution of all data? For advanced techniques, software and processing details should be provided. For example, how are turbulent fluxes being computed? How is the mixing height derived? What are the scanning patterns for the lidar sensors? . . .

Some observations are shown, but it is not made very clear why the specific data are presented. A few interesting key results would be more useful to demonstrate the most important capabilities of the measurement network.

P2, l14: Why is solar radiation decreased? Check your argument: less solar radiation does not result in increased sensible heat flux and storage.

P2, l16: Check your argument: is there really more convection if synoptic winds are stronger?

P2, l20: Define what you mean by 'user-specific'

P2, l27: You are listing BUBBLE, TOMACS and SUIMON campaigns. State clearly why you are describing these three examples.

P3, l2: What is meant by 'cumulus'?

P3, l8: reword the sentence. Turbulent fluxes result from surface-atmosphere interactions. Not quite clear what is meant by '. . . fluxes alter boundary layer structure through boundary layer processes'.

P3, l14: use different word. Weather can not be 'unreasonable'

P4, l31: Was 'SKP' introduced?

P5, l30: reword. EB are not installed on the ground or rooftop but rather the towers.

P6, l5: How do you measure thermal conductivity and heat capacity? Are these estimates representative of later areas?

P6, l10: How are the turbulent fluxes calculated? What QAQC is applied?

P6, l34: Might be good to mention the need for mosquito monitoring earlier in introduction.

P6, l12: Define 'urban structure'. What are you referring to with 'communication'? Are do you determine urban structure, surface cover, metabolism, communication etc. Scales, methods, underlying data sources...?

P6, l13: More information is needed on sensor calibration and inter-comparison.

P7, l16: More details required on OSSE. What is being done and why.
* * *

---

## Referee Comment (RC2) · Anonymous Referee #2 · 5 Dec 2016

The manuscript describes a new system combining various measurement techniques to be assimilated into a high resolution model of the Seoul metropolitan area. The design of the network is very challenging and, once operational, should significantly help the now- and fore-casting of air-pollution events in the metropolitan area of Seoul.

Meanwhile, to me, the absence of (at least) case studies or results illustrating the ability of such an integrated system within an urban area to improve now- and fore-casting makes the paper unpublishable in AMT.

I fully agree with the first reviewer when he says that "The paper does not fit the scope of the AMT journal, however, the measurement network is very impressive and addresses the great need for detailed observations in urban areas, I would encourage

the authors to submit the paper elsewhere."
* * *

---

## Author Comment (AC1) · 25 Jan 2017

Manuscript: "High-Resolution Urban Observation Network for a User-Specific Meteorological Information Service in the Seoul Metropolitan Area, Korea" by Moon-Soo Park et al.

Response to the Anonymous Referee #1

Authors gratefully thank the reviewer for his/her thorough review and valuable comments which contributed to improve the manuscript. Reviewer's all comments except for journal scope are responded. According to the reviewers' suggestion, authors will add more detailed explanations on the observation system, and two case studies to demonstrate the usefulness and applicability of the UMS-Seoul in Chapter 5. Reviewer's comments are marked in black, while authors' responses are marked in blue.

On journal scope

⇨ This manuscript had been submitted to AMT by accepting the ACP (Atmospheric Chemistry and Physics) Executive Editor's suggestion. Authors think that this topic is included in the meteorological measurement platform, one of AMT journal's main scopes. Design of an intensive & integrated meteorological observation network for a special purpose is as important as development or improvement of each measurement instrument.

1-1. Text, maps, pictures should include much more details on the observing systems. For example, for all sensors, manufacturer and model should be listed. How do sensors inter-compare? How do you ensure absolute calibration? What is the accuracy of different measurements? What is the temporal resolution of all data?

⇨ More detailed descriptions on the observation systems will be added. (1) Manufacturer, model, measurement range, and accuracy or sensitivity of sensors deployed on the surface energy balance system will be listed in Table 6. (2) Most sensors are certified by manufacturer before shipping, and by Korea Meteorological Promotion Agency (sponsored by Korea Meteorological Administration) before installation. Performance certificates have been issued and renewed at every 3 years. (3) A $H_2O$ and $CO_2$ infrared gas analyzer is

calibrated every 6 month according to procedure suggested in the manufacturer's manual. (4) Performances for surface-based remote sensing instruments are also certified by their own manufacturers before shipping. Also vertical profiles obtained by these instruments are compared with boundary-layer structures obtained by sonde before installation and during the intensive sonde observation campaign period every year. (5) Manufacturer, model, and temporal resolution for each instrument are added in Table 4.

1-2. For advanced techniques, software and processing details should be provided. For example, how are turbulent fluxes being computed? How is the mixing height derived? What are the scanning patterns for the lidar sensors?

⇨ Related descriptions will be added. (1) CR3000 for data logging and LoggerNet for operating software manufactured by Campbell Scientific In. are used for surface energy balance system. (2) Quality check algorithms for meteorological variables and flux data are developed according to the standard procedures. The quality check for meteorological variables are developed by the comparison UMS-Seoul's algorithm with those developed by KMA, WMO, and previous studies (Chae et al., 2014). Surface fluxes are computed from 10 Hz raw data using the following procedure: physical limit check, detection and removal of spike data (Vickers and Mahrt, 1997), computation of the vertical flux with a 30-minute block average (Kwon et al., 2014), Webb-Pearmann-Leuning correction (Webb et al., 1980; Leuning, 2007). (3) Wind lidar applies the Doppler beam swinging scanning technology to determine the wind speed and direction at a given height (Werner, 2005). (4) Mixing height is determined as a height with a minimum gradient of backscattered coefficients obtained by a ceilometer or an aerosol lidar (Eresmaa et al., 2006) or with the steepest decrease of wind speed variance obtained by a wind lidar (Emeis et al., 2008).

2. Some observations are shown, but it is not made very clear why the specific data are presented. A few interesting key results would be more useful to demonstrate the most important capabilities of the measurement network.

⇨ Authors will add two case studies using the observation data in Chapter 5. One is for the spatial distribution of surface meteorology and temporal evolution of urban boundary-layer structures during the 3 consecutive days in spring, the other is for finding the road sections vulnerable to road wetness and ice using the surface temperature and status on a roadway obtained by a mobile road weather vehicle.

3. P2, L14: Why is solar radiation decreased? Check your argument:

⇨ More detailed descriptions on solar radiation decrease and sensible heat flux/heat storage increase will be inserted. (1) Solar radiation in urban area is smaller than that in rural area due to visibility reduction by anthropogenic aerosol and air pollutants (Peterson et al., 1978; Robaa, 2009). (2) Sensible heat flux and heat storage increase is due to anthropogenic heat release from urban surface and a decrease in latent heat flux.

4. P2, L16: Check your argument: Is there really more convection if synoptic winds are stronger?

⇨ The sentence will be rewritten. Actually, it means that when the synoptic wind becomes strong, the most precipitation area with the strong upward motion moves to more downwind.

5. P2, L20: Define what you mean by 'user-specific'.

⇨ 'User-specific' means 'customized for users' demands'. For example, meteorological variable, and spatial and temporal resolution of the variables for flash-flood is quite different from those for road weather information service.

6. P2, L27: You are listing BUBBLE, TOMACS, and SUIMON campaigns. State clearly why you are describing these three examples.

⇨ Related sentences will be rewritten. Meteorological and observation variables, spatial resolution for each instrument in the network, and temporal resolution for each variable are determined according to their own purposes: Surface meteorology and service-oriented observation are enough for real-time information service such as the New York mesonet (http://www.nysmesonet.org); surface energy balance and vertical profiles are needed for high-quality and high-resolution forecast such as the Basel Urban Boundary Layer Experiment (Rotach et al., 2005), and the Shanghai Urban Integrated Meteorological Observation Network (Tan et al., 2015); radars are good for real-time service for severe weather and short-term forecast such as the Tokyo Metropolitan Area Convection Study for Extreme Weather Resilience (Nakatani et al., 2013; 2015).

7. P3, L2: What is meant by 'cumulus'?

⇨ More detailed description on the physics schemes including 'cumulus' will be added. Microphysics scheme deals with the interactions among water vapor, cloud water, cloud ice, rain drop, snow, and graupel. Cumulus scheme deals with the updraft, downdraft, entrainment,

and detrainment in cloud. Radiation scheme deals with the absorption, emission, scattering, reflection, transpiration in the atmosphere for radiative energy. Surface scheme deals with the surface energy balance and energy/moisture transfer between the surface and ground. Atmospheric boundary layer scheme deals with the energy and moisture transfer between the surface and the atmospheric boundary layer.

8. P3, L8: Reword the sentence. Turbulent fluxes result from surface-atmosphere interactions.

   ⇨ The sentence will be rewritten. The modified sensible and latent heat fluxes change the boundary-layer structure through the energy and moisture interactions among the surface, underlying ground, and overlying atmosphere.

9. P3, L14: Use different word. Weather cannot be unreasonalble.

   ⇨ 'Unreasonable' will be changed to 'extremely'.

10. P4, L31: Was 'SKP' introduced?

   ⇨ More description will be added. 'SKP' is a subsidiary company of SK, the largest telecommunication company in Korea.

11. P5, L30: Reword. EB are not installed on the ground or rooftop but rather the towers.

   ⇨ The sentence will be rewritten. All 14 surface energy balance observation system are deployed on the tower installed on the ground, or on a rooftop of building surrounded by the representative surface land cover.

12. P6, L5: How do you measure thermal conductivity and heat capacity? Are these estimates representative of later areas?

   ⇨ More detailed description will be added. Surface thermal conductivity, heat capacity, and emissivity are estimated and verified by comparison the surface temperature determined by energy balance and heat transfer models with the observed surface temperature. And surface roughness length and displacement length are determined using the urban morphology data such as mean building height, frontal area density, and plane area density obtained by geographical information system.

13. P6, L10: How are the turbulent fluxes calculated? What QAQC is applied?

   ⇨ More description will be added. Basic quality checks for meteorological variables include a

missing check, a physical limit check, a climate range check, and a spike removal (Chae et al., 2014). Surface fluxes are computed from 10 Hz raw data using eh following procedure: (1) physical limit check, (2) detection and removal of spike data (Vickers and Mahrt, 1997), (3) computation of the vertical flux with a 30-minute block-average (Kwon et al., 2014), (4) Webb-Pearmann-Leuning correction (Webb et al., 1980; Leuning, 2007).

⇨ Because each instrument has its own data characteristics with its own format, temporal resolution, and vertical resolution, so it should have its own quality check algorithm. For example, wind lidar has the following quality check procedure: carriage-to-noise check, data availability check, and vertical gradient of horizontal wind check in addition to the basic missing check (Park and Choi, 2016).

14. P6, L34: Might be good to mention the need for mosquito monitoring earlier in introduction?

⇨ Mosquito entity is one of ecological service. Ecology will be mentioned in introduction.

15. P6, L12: Define 'urban structure'. What are you referring to with 'communication'? Are do you determine urban structure, surface cover, metabolism, communication etc., Scale, method, underlying data sources?

⇨ More detailed description on the urban structure and telecommunication will be added. Also, communication is changed to telecommunication to avoid its ambiguity in the revised manuscript. Urban structure includes spaces between buildings, building density, street widths, tree height, tree species, etc. (Muller et al., 2013). And telecommunication includes telecommunication type, name, password, backup, network owner, contact, etc.

16. P6, L13: More information is needed on sensor calibration and inter-comparison.

⇨ More information will be added. Meteorological sensors such as air temperature, relative humidity, air pressure, wind speed, wind direction, solar radiation, and precipitation are certified by the Korea Meteorological Industry Promotion Agency before installation and in every 3 year. Infrared $H_2O/CO_2$ gas analyzer is calibrated every 6 months according the manual (http://s.campbellsci.com/documents/af/manual/ec150.pdf).

17. P7, L16: More details required on OSSE. What is being done and why?

⇨ The sentences will be rewritten. The station locations are optimized using the analysis of observing system simulation experiment (OSSE), a model-base experiment for the purpose of

assessing the potential impact of the would-be observation station for any instrument and/or sensor (Zhang and Pu, 2010), in order to give the meteorological information service more stable. According the OSSE results, some stations or instruments may be added, removed, or moved to other locations.

**Table 6 and the following chapter will be added in the revised manuscript.**

Table 6 Manufacturer, model, measurement range, and accuracy or sensitivity of sensors deployed on the surface energy balance system.

| Sensor | Manufacturer (Model) | Measurement range | Accuracy |
|---|---|---|---|
| 3D sonic anemometer | CSI[*] (CSAT3B) | u & v: ±60 m/s, w: ±8 m/s, Ts: -50~60℃ | u & v < ±0.08 m/s, w < ±0.04 m/s |
| $CO_2$ and $H_2O$ open-path gas analyzer and 3D sonic anemometer | CSI/Li-Cor (EC150/ CSAT3A) | $CO_2$: 0~1,798 mg/m$^3$ at 25℃, 1 atm $H_2O$: 0~52.9 g/m$^3$ at 25℃, 1 atm **3D Sonic anemometer** u & v: ±60 m/s, w: ±8 m/s, Ts: -50~60℃ | **$CO_2$** Precision (RMS): 0.2 mg/m$^3$ Zero drift: ±0.55 mg/m$^3$/℃ Gain drift: ±0.1% of reading/℃ **$H_2O$** Precision (RMS): 0.004 g/m$^3$ Zero drift: ±0.037 g/m$^3$/℃ Gain drift: ±0.3% of reading/℃ **3D Sonic anemometer** u & v < ±0.08 m/s, w < ±0.04 m/s |
| Net radiometer | Kipp & Zonen (CNR4) | **Maximum irradiance**: 4000 W/m$^2$ (short-wave); 2000 W/m$^2$ (long-wave) **Spectral range**: 300~2800 nm (short-wave); 4500~42000 nm (long-wave) | **Expected daily uncertainly**: < 2% (short-wave); < 10% (long-wave) **Sensitivity**: 7~20 µV/W/m$^2$ (short-wave); 5~10 µV/W/m² (long-wave) |
| Temperature and relative humidity Probe | CSI/ Vaisala (HMP155A) | **Temperature**: -80~60℃ **Relative Humidity**: 0~100% | **Temperature**: ±0.3℃ **Relative Humidity**: ±1% at 15~25℃, 0~90%, ±1.7% at 15~25℃, 90~100% |
| Wind vane | Vector Instruments (W300P/A100M) | 0~360° | ±3° |
| 3-cup anemometer | | 0~75 m/s | 0.1 m/s at 0.1~10 m/s; 1% at 10~55 m/s; 2% at 55~75 m/s |
| Barometric pressure | CSI/ Vaisala (CS106) | 500~1100 hPa | ±0.3 hPa at 20℃; ±0.6 hPa at 0~40℃; ±1.0 hPa at -20~45℃ |
| Infrared surface temperature | CSI/ Apogee (SI-111) | -40~70℃ | ±0.5℃ at -40~70℃ |
| Precipitation gauge | Wedaen (WDSA-205) | 0.5 mm per 1 tip | ±3 mm at 150 mm/hour |
| Soil heat flux plate | CSI/Hukseflux (HFP01) | -2000~2000 W/m$^2$ | Sensitivity: 50µVW$^{-1}$m$^2$ -15~5% in most common soil |
| Soil water content | CSI (CS655) | **Soil temperature**: -10~70℃ **Volumetric water content**: 5~50% | **Soil temperature**: ±0.5℃ **Volumetric water content**: ±3% |
| Soil & water Temperature | CSI (107) | -35~50℃ | ±1.0℃ |
| Propeller-type wind vane | RM Young (05103) | **Wind speed**: 0~100 m/s **Wind direction**: 0~360° | **Wind speed**: ±0.3 m/s **Wind direction**: ±3° |

[*] CSI: Campbell Scientific Inc.

**5. Case Studies**

In order to demonstrate the usefulness of UMS-Seoul and applicability to the meteorological information service customized users' demands, two case studies are conducted: One is for the spatial distribution of surface meteorology and the evolution of urban boundary layer structures during the 3 consecutive days in spring, the other is for finding the road sections vulnerable to road wetness and ice using the surface temperature and status on a roadway obtained by a mobile road weather vehicle.

**5.1. Case Study I: spring zonal anticyclone event**

Surface meteorology and atmospheric boundary layer structures are investigated for the period from 18 to 20 May 2016. During this period, a zonal anticyclone in northern Japan blocks an eastward moving weather system. Thus, a jet stream at 300 hPa level is divided into higher and lower latitude directions over eastern China. As a result, Seoul Metropolitan Area shows a fine weather at the edge of high pressure system. A short-lived and small thermal low pressure system driven by thermal difference between continent and sea is developed in the afternoon and disappears in the evening on 19 and 20 May 2016.

Figure 6 shows a horizontal distribution of air temperature obtained by SKP surface meteorological observation system at 0600 LST, 1200 LST, 1800 LST, and 2400 LST on 18 May 2016. At 0600 LST and 2400 LST when there is no thermal heating, western sites show a relatively high temperature, while eastern sites show a relatively low temperature. On the other hand, at 1200 LST and 1800 LST, eastern sites show a relatively high temperature, while western sites show a relatively low temperature. This temperature difference implies the existence of local circulation such as land and sea breeze, and mountain valley circulation. Urban sites have higher temperature than the surrounding sites throughout the period, which is mainly due to heat capacity difference between the urban and the rural. Temperature difference between two land coves, that is urban heat island effect, becomes stronger during the night.

Figure 7 shows the time series of surface meteorological variables obtained at every 1 minute by a surface energy balance system installed at the Jungnang station for the period from 0000 LST 18 to 2400 LST 20 May 2016. During this period, it was so clear that daily cloud cover was recorded as 0.5/10, 0.0/10, and 0.6/10 in 18, 19, and 20 May 2016, respectively. Air pressure minimum is occurred at around 1500-1800 LST every day (Figs. 7a, b, d, e), which is accompanied by a wind speed, direction, and vapor pressure change. That is to say, after low air pressure passes the station, wind direction is abruptly changed to northwesterly, air temperature drops down by 1.8 ℃ (3.5 ℃), and

vapor pressure jumps up by 2.6 hPa (8.0 hPa) on 19 (20) May. Diurnal variation of wind speed and direction shows that the station is affected by local circulations: northeasterly winds are dominant at night, while other directional winds are dominant in a day (Figs. 7c and d). Net radiation is negative at night and positive during the day, the variation of which shows that there are few clouds in this period except for in the afternoon 20 May (Fig. 7f).

Figure 8 shows the backscattering coefficient observed by a ceilometer and vertical profile of wind observed by a wind lidar at the Jungnang station for the period from 0000 LST 18 to 2400 LST 20 May 2016. Atmospheric boundary-layer structures defined by a backscattering coefficient are perfectly coincident with those defined by a wind: (1) Backscattering coefficient shows that there are two distinct layers before 1000 LST 18 May; the lower layer with a maximum height of 400 m contains thick backscattering aerosols, while the upper layer has less dense to 1.2 km. Wind profile also shows that two layers have different origins: Easterly winds are dominant at lower layer, while westerly winds are dominant at upper layer; (2) Convective atmospheric boundary layer defined by a backscattering coefficient evolves during the day on 3 consecutive days. Correspondingly, winds become irregular throughout the same layer; (3) Residual layer with a high backscattering coefficient around 2 km high at night move downward slowly to the next morning time, and combine with the evolving convective boundary layer at noon in 19 May. Southerly winds are in upper residual layer, while northerly or north-easterly winds are in lower stable boundary layer; High backscattering coefficient zones at from 500 m to 1500 m high on 0300 to 09 LST and at height than 1000 m on near 1800 LST 20 May are exactly corresponded to the wind convergence zone. Potential temperature and mixing ratio profiles obtained by a microwave radiometer also supports the similar atmospheric boundary-layer structures (Fig. 9). To be concluded, surface meteorology and vertical profiles of meteorological variables observed from the UMS-Seoul will be very helpful to produce the high-quality and high-resolution meteorological field in the SMA.

5.2. Case study II: mobile road weather vehicle

Figure 10 shows an example of the road surface temperature service on a roadway route in Seoul Metropolitan Area observed by a mobile road weather vehicle with the road material, structure and elevation for the period from 1000 to 1440 LST on 2 December 2016. The road material is classified as asphalt (72.2 %) and concrete (27.8 %), while road structure is classified as over-ground (86.2 %), bridge (10.4 %), underpass (0.6 %), and tunnel (2.8 %) roads (Table 7). The road elevation is observed by the global positioning system sensor. The surface temperature is found to be related to the road elevation, surface material, road structure, sky-view, and horizontal distribution of surface land

use. The surface temperature over concrete-road is lower than that over asphalt-road due to the difference of albedo and diffusivity (Fig. 11). While that over bridge is higher than that over an over-ground due to the difference of thermal heat capacity and heat transfer processes. Using these data, the roadway sections vulnerable to road wetness and icing on the highways and major principal roads can be determined (Fig. 12). These vulnerabilities will be applicable to give the alarm or advisory to drivers. Also, these data will be applicable to improve the road surface temperature and status prediction system (Park et al., 2014).

Table 7 Coverage fraction for road material and structure on the roadway route observed during 1000 and 1440 LST on 2 December 2016.

| Material | Structure | Coverage (%) |
|---|---|---|
| Asphalt | Over-ground (AG) | 67.18 |
| | Bridge (AB) | 3.77 |
| | Underpass (AU) | 0.57 |
| | Tunnel (AT) | 0.67 |
| Concrete | Over-ground (CG) | 19.04 |
| | Bridge (CB) | 6.59 |
| | Underpass (CU) | 0.00 |
| | Tunnel (CT) | 2.18 |
| Total | | 100.00 |

[Figure]

Figure 6: Horizontal distribution of air temperature obtained by SKP surface meteorological observation system at (a) 0600 LST, (b) 1200 LST, (c) 1800 LST, and (d) 2400 LST 18 May 2016 in the Seoul Metropolitan Area. Dotted rectangle denotes the high-populated region in Figure 1b.

[Figure]

Figure 7: Time series of (a) air temperature, (b) vapor pressure, (c) wind speed, (d) wind direction, (e) air pressure, and (f) net radiation observed at the Jungnang station for the period from 18 to 20 May 2016.

[Figure]

Figure 8: Time-height cross sections of (a) backscattering coefficient obtained by a ceilometer, and (b) wind speed and direction obtained by a wind lidar at the Jungnang station for the period from 0000 LST 18 to 0000 LST 21 2016.

[Figure]

Figure 9: Time-height cross sections of (a) (a) potential temperature and (b) mixing ratio obtained by a microwave radiometer at the Jungnang station for the period from 0000 LST 18 to 0000 LST 21 May 2016.

[Figure]

Figure 10: (a) Road surface temperature, (b) road surface material and structure, and (c) elevation on the roadway obtained by a mobile road weather vehicle in the Seoul Metropolitan Area for the period from 1000 LST to 1440 LST 2 December 2016.

[Figure]

Figure 11: Boxplot of observed road surface temperatures according to road material and structure on the roadway obtained by a mobile road weather vehicle in the Seoul Metropolitan Area for the period from 1000 LST to 1440 LST 2 December 2016. First character stands for road material type: A does for Asphalt, C for concrete. Second character denotes road structure type: G for over-ground, B for bridge, U for underpass, and T for tunnel.

[Figure]

Figure 12: Detailed surface temperature and material & structure with a satellite image at road sections (a) from 126.88E to 126.98E and (b) from 126.68E to 126.76E in Figure 10. First character stands for road material type: A does for Asphalt, C for concrete. Second character denotes road structure type: G for over-ground, B for bridge, U for underpass, and T for tunnel.